# Benefits of Polyphenols and Methylxanthines from Cocoa Beans on Dietary Metabolic Disorders

**DOI:** 10.3390/foods10092049

**Published:** 2021-08-31

**Authors:** Elodie Jean-Marie, Didier Bereau, Jean-Charles Robinson

**Affiliations:** Laboratoire COVAPAM, UMR Qualisud, Université de Guyane, 97300 Cayenne, France; elodie.jean-marie@univ-guyane.fr (E.J.-M.); didier.bereau@univ-guyane.fr (D.B.)

**Keywords:** cocoa, polyphenol, methylxanthines, intestinal immunity, microbiota, diabetes

## Abstract

*Theobroma cacao* L. is an ancestral cultivated plant which has been consumed by various populations throughout history. Cocoa beans are the basic material occurring in the most consumed product in the world, namely chocolate. Their composition includes polyphenols, methylxanthines, lipids and other compounds that may vary qualitatively and quantitatively according to criteria such as variety or culture area. Polyphenols and methylxanthines are known as being responsible for many health benefits, particularly by preventing cardiovascular and neurodegenerative diseases. Recent studies emphasized their positive role in dietary metabolic disorders, such as diabetes and weight gain. After a brief presentation of cocoa bean, this review provides an overview of recent research activities highlighting promising strategies which modulated and prevented gastro-intestinal metabolism dysfunctions.

## 1. Introduction

*Theobroma cacao* Linnaeus 1753 is a food plant that has been domesticated by Mesoamericans and propagated to European countries in 1502 [1]. Originating from neotropical rainforests, its fruits are called “pods” and contain “cocoa beans” with a surrounding muscillagineous sweet pulp. To become one of the most widely consumed products in the world, namely chocolate, cocoa seeds must undergo several processes such as fermentation, drying and roasting [2]. Consumed for its flavor, raw cocoa and cocoa-derived products revealed human health benefits as protective against cardiovascular and neurodegenerative diseases, antioxidant and anti-inflammatory activities [3,4]. Bioactive compounds, particularly polyphenols and methylxanthines, are thought to be responsible for these properties [5,6]. Nowadays, dietary metabolic disorders, such as diabetes, are on the rise, and recent in vitro and in vivo studies on cocoa compounds revealed preventive actions such as anti-diabetic and anti-obesity activities [7,8,9]. In addition, cocoa modulated the profile of intestinal microbiota, which can provide health benefits to the host [10]. The present review focuses on cocoa botany, its polyphenol and methylxanthine compositions and its effects on dietary metabolic and gastrointestinal disorders. In addition, we focus on the impact of these compounds on the intestinal microbiota and possible interferences with metabolic disorders.

## 2. Taxonomy

According to its ancestral origin, cocoa is called the drink of God (*Theo* = God, *Broma* = drink) and belongs to the *Theobroma cacao* L. plant species. Its discovery is attributed to Olmec population in 1500–400 BC. Then, it was cultivated by Mesoamericans (600 BC) and Aztecs (400 AC) because of its divine origin and was turned into “ancestral” chocolate [11].

Cocoa belongs to Malvales order and Malvaceae family which is divided into 6 major groups containing 22 species [12,13]. Based on morphological/botanical parameters, in 1882, Morris classified cocoa into two subspecies, Criollo and Forastero. Criollo was the first domesticated cocoa and was widely grown in Central America and Northern South America [14]. Native to the lower Amazonia, the first Forastero was cultivated in Brazil and Venezuela, but a particular morphological type called Amelonado was introduced into Africa [14]. In 1901, Preuss proposed a new morphological classification of cocoa in three subspecies: Criollo, Forastero and Trinitario. Indeed, the latter would be found in Trinidad after a spontaneous hybridization between Criollo (already cultivated) and Forastero (introduced in 1727) [15,16].

Advances in genetic engineering have deepened knowledge of cocoa genome and varieties. Indeed, some distinctions based solely on morpho-geographical criteria were supported. For example, a genetic distinction was observed between the High Amazonians and Low Amazonians Forasteros, Criollos and Trinitarios by ribosomal DNA probes or RAPD markers [17,18]. Similarly, Trinitario differs from Forastero in allozyme data [19]. However, in 2008, Motamayor and collaborators highlighted incoherence in the classification of the three original varieties and indicated that they were 10 distinct and specific genetic groups: Maranon, Curaray, Criollo, Iquitos, Nanay, Contamana, Amelonado, Purus, Nacional and Guiana [20].

Nowadays, the botanical classification of cocoa is still based on the old morphological classification “Criollo, Forastero and Trinitario”, as well as on the genetic classification of Motamayor and collaborators. Cocoa can also be identified by its “type”, a parameter determined by the shape of the pod. This type may sometimes have the same name as the genetic group. For example, a cocoa pod of type Amelonado can be genetically Amelonado or Forastero. In addition, the cocoas are often described according to the hybridizations from which they are derived and listed in genetic banks. For example, cocoa trees grouped by Pound from 1933 to 1935 are grouped under the “Imperial College Selections (ICS)” and are referred to as ICS 1, 3 or 999 [21].

## 3. Bean Morphology

The cocoa fruit is a lengthened berry, called a pod, which is 10–32 cm long and whose appearance plays a crucial role in the identification of varieties and populations. Indeed, its surface that is characterized by ten ridges can be smooth to warty. Bottle neck, shoulder and apex can be pronounced. Color can vary from very light green to dark green, from yellow/red to deep purple [22]. A pod contains between 30 and 60 cocoa beans which are surrounded by plentiful mucilage [1].

Beans have variable shapes and sizes, with a 25 mm long and 8 mm thick flattened aspect. Bean is characterized by (1) an almond that contains two cotyledons and a small embryo, then (2) a shell which surrounds them. Cotyledons are storage organs containing the necessary resources for the development of the seedling. They also contain the first two leaves of the plant when the seed germinates [23]. In the embryo, the hypocotyl represents the middle part of the stem bellow cotyledons, and the radicle represents the lower part of the embryo which becomes the roots. The Figure 1 illustrates the morphology of the cocoa bean.

In the Criollo variety, the pod is characterized by fine red or yellow walls and smooth rectangle shape; also, seeds have a particular flavor. Indeed, this variety is catalogued as one having the finest cocoa flavor, while Forastero is considered by its quality as bulk. Trinitario can be considered as bulk or fine cocoa. Nacional, a subgroup of Forastero from Ecuador, is defined as fine cocoa [25].

## 4. Bean Composition

Cocoa is mainly valued in the food sector and can be used to make chocolate, cocoa butter, nibs, or cocoa powder. There is different distribution of the compound in each part of the bean. The almond is distinguished by its high fat content while the shell is a protein and fiber source. Their composition is shown in Table 1.

Polyphenols and Methylxanthines are bioactive compounds thought to be responsible for several human health benefit of cocoa [5,6]. Their chemical structures are presented in Figure 2.

### 4.1. Polyphenols

In cocoa, dry whole bean weight is mainly due to polyphenols (10–18%) [29]. They represent one of the most widespread compounds families in plant kingdom, with more than 8,000 known structures [30]. Polyphenols are secondary metabolites and are thought to be responsible in cocoa for the bitterness and astringency of the seeds and chocolate [23]. Two main groups of polyphenols can be distinguished: (1) monomeric and polymeric flavanols (nearly 37% and 58% of the total polyphenols of the bean, respectively), and (2) anthocyanins (4% of the total polyphenols of the bean) [5].

Flavanols are the most present flavonoids in food plants and are major in cocoa beans. Their amount would be higher in cocoa than teas and wine [31]. (−)-Epicatechin would be the main flavanol with a proportion of 35 %, while other subgroups such as (+)-catechin, (+)-gallocatechin, and (−)-epigallocatechin would be in smaller amounts. The main procyanidins compounds are dimeric, trimeric (or with a higher degree of polymerization) flavanols such as B1, B2, B3, B4, B5, C1 and D [5]. With a 4% total polyphenol content in cacao seeds, anthocyanins are represented by cyanidine-3-α-L-arabinoside and cyanidine-3-β-D-galactoside. They are commonly found associated with glucose, arabinose, or galactose in cocoa [5].

In minor quantities, flavonols or flavones are found as other polyphenols. Aglycone and glycosylated quercetins accounted for most part of flavonols, including quercetin-3-O-glucoside, (isoquercetin), quercetin-3-O-arabinoside, quercetin-3-O-galactoside (hyperoside) and quercetin-3-O-glucuronide (Andres-Lacueva et al., 2008) [31]. In the case of flavones, cocoa seed contains apigenin and luteolin in their aglycone or glycosylated form including apigenin-8-C-glucoside (vitexin), apigenin-6-C-glucoside (isovitexin), luteolin-7-O-glucoside, luteolin-6-C-glucoside, and luteolin-8-C-glucoside [32]. Flavanones were also found, such as naringenin and naringenin-7-O-glucoside (prurin) [28]. Moreover, phenolic acids, subdivided in hydroxycinnamic acids such as chlorogenic, syringic, caffeic, p-coumaric acids, but also hydroxybenzoic acids such as vanillic and gallic acids [28,33] can be found in cocoa. Trans-resveratrol and trans-piceid, resveratrol derivates, were also found in cocoa [34].

Cocoa polyphenol content is variable and depends on many parameters. By comparing high Amazonia Forastero, Nacional, Criollo and Trinitario cocoas, differences in phenolic contents suggested a variety influence. Criollo would have little or no anthocyanidins, while Forastero would have a higher (−)-epicatechin content [35]. Geographical origin could also have an influence, and this was highlighted in a polyphenolic fingerprinting of cocoa from various origins. This study showed quantitative and qualitative differences between Indonesia and Ivory Coast cocoas [36]. Similarly, growing culture modulates the phenol content because soil type, altitude, sun exposure could have significative impacts [37,38]. Maturity is also an important parameter because during development, polyphenols gradually accumulate in storage cells [39]. From maturity stage 1 (purple-red pods) to stage 4 (orange pods), catechin contents increased from 6.39 ± 0.02 to 8.04 ± 0.24 g/100 g dry matter [40].

Furthermore, manufacturing and processing could play a key role in the final quality and quantity of polyphenols remaining in cocoa-derived products [41]. For example, after drying, total soluble polyphenols in defatted cocoa seeds decreased from 20 to 6% [3]. Fermentation induced a 70% decrease in the phenolic content and a 90% decrease of (−)-epicatechin content [42]. Acidic conditions, heat and enzymatic activities are thought to be responsible for the decrease in polyphenols and methylxanthines. Fermentation is characterized by a set of enzymatic reactions orchestrated by certain enzymes such as polyphenol oxidase and glycosidase. For example, polyphenol oxidase converts mainly epicatechins and anthocyanidins into quinones, reactive molecules capable of complexing with proteins and other polyphenols [43]. Fermentation could also lead to the emergence of new compounds. Indeed, this phenomenon revealed intra- and inter-related associations between the B-cycle of flavanols. This is the case for new dimeric compounds “F3” and “F4” which are thought to derive from associations between epicatechins only and epicatechins and cyanidins, respectively [44].

To determine the fraction of polyphenols in cocoa, several analytic methods are applied, such as the following:Colorimetric/Spectrophotometric (UV-Vis) methods [4,45,46,47,48,49,50]:
οTotal polyphenol content by the Folin–Ciocalteu methodοTotal anthocyanidins content by the analysis of pH variationοTotal flavonoids content by AlCl_3_ methodοTotal proanthocyanidins by Vanillin, DMACA, butanol-HCl or bovine serum albumin (BSA) methodsChromatographic methods [51,52]:
οThin layer chromatographyοHigh performance liquid chromatography (Identification with or without mass spectrometry detector/Determination of the polymerization degree of tannin by thiolysis such as phloroglucinolysis)

### 4.2. Methylxanthines

In cocoa seed, dominant purine alkaloids are theobromine (3,7-dimethylxanthine) and caffeine (1,3,7-trimethylxanthine) [53]. They are also secondary metabolites which are responsible for astringency and bitter taste of cocoa and derived products. In the raw cocoa seed, theobromine is predominant and represents nearly 2.7% and 0.7% in almond and shell dry weight respectively, while caffeine represents nearly 0.8 and 0.6% [26]. Theophylline (1,3-dimethylxanthine) is found as traces in cocoa [53]. These three compounds are located in the same storage cells as polyphenols, which are recognized by their single large vacuole [54].

The cocoa methylxanthine content is linked to various parameters such as genetic, geographic or maturity factors. Forastero and Trinitario showed similar theobromine levels (about 1%) but distinct caffeine levels (0.15% and 0.30%, respectively). Criollo was distinguished from the others by a lower theobromine content (0.77%) and a higher caffeine content (0.53%) [55]. Geographic influence was noticed by comparison of two Forastero hybrids (SR160 and PH16) issued from two regions in Bahia (Brazil) which showed different caffeine contents: 132 ± 70 mg/100g dry matter for SR160 and 365 ± 46 mg/100g MS for PH16 [56]. The maturity stage also influences the methylxanthine content as there was a 70.5% and 74.2% increase in theobromine and caffeine, respectively, between the immature and mature stage [57].

To determine the fraction of methylxanthines in cocoa several analytic methods are applied, such as the following [58,59]:Spectrophotometric methods such as AOAC Micro Bailey–Andrew method and Morton–Stubb methodThin-layer chromatographyGas ChromatographyLC (Liquid chromatography) or HPLC (High performance liquid chromatography) with or without MS (mass spectrometry detector) for identificationsCapillary electrophoresis

## 5. Health Benefits of Polyphenols and Methylxanthines from Cocoa

### 5.1. Cocoa and Intestinal Inflammation

Immune system is distinguished by two immune responses: the innate and the adaptative immune system (IS and AS). The first one is managed by a poor specific phenomenon named “inflammation”. Indeed, during inflammation, IS cells such as macrophages have structural motifs in their surfaces which can recognize large set of pathogens. For example, macrophages recognized bacterial endotoxins via lipopolysaccharide (LPS) receptors or several virus double-stranded DNA [60,61]. Surface receptors include toll-like receptors (TLRs), which are specific to microbial components. For example, TLR-4 is specific for bacterial LPS, whereas TLR-3, TLR-7/8 and TLR-9 are specific to viral nucleic acids [60,61]. Once activated, various inflammatory mechanisms could induce the synthesis of proteins, enzymes, or activate a huge amount of transcription factors. They include the nuclear factor κB (NF-κB), which induces various cytokines and interferon response factor-3 (IRF-3). Then, they stimulate type 1-interferons (IFN-α, IFN-ω…) and cytokines production blocking viral replication [62,63]. The second immune system named AS is characterized by a selective memory and is distinguished by humoral (B-cells and antibodies) and cellular (T-cells and polynuclear neutrophiles phagocytosis) immunities [62].

If inflammation is a beneficial and necessary process, when it becomes chronic and able to attack its own host, it becomes a problem. One of its disadvantages is the fact that inflammation and oxidative stress are two closely related phenomena. Indeed, oxidative stress (condition of imbalance between the production and the destruction of oxidative compounds that become toxics to cells) may be a consequence of inflammation. Macrophages and polymorphonuclear neutrophils are important sources of oxidizing compounds called reactive oxygen species (ROS) through a membrane enzyme called NADPH oxidase that is activated after a stress trigger by inflammatory process. To protect their integrity, the cells induced the release of proteolytic enzymes that produce superoxide anion O_2_^−•^. This free radical could lead to the production of other toxic radicals as hydrogen peroxide H_2_O_2_ (after dismutation), hypochlorous acid HClO, hydroxyl radical OH^•^ and nitric oxide NO [64]. Paradoxically, oxidative stress may also be the cause of inflammation. Indeed, H_2_O_2_ and OH^•^ can lead to the production of proinflammatory cytokines such as IL-6 and TNF-α by the modulation of NFκB, while they are themselves modulated by this factor [65,66]. There was a correlation between antioxidant potential of polyphenols and the molecular effects of anti-inflammatory and chemo-protective activities. This may be due to their ability to modulate signaling cascades and gene expressions involved in cell-proliferation, apoptosis and chronic inflammation [67], but also to their ability to scavenge free radicals [68].

Intestinal mucosa is composed of epithelial cells lines which act as a permeable barrier and regulate intestinal immune system facing dietary pathogens or microbiota [69]. Oxysterol is derived from dietary cholesterol which causes dysfunction and epithelium permeability damages on intestinal mucosa by inducing inflammation and ROS overproduction. Cocoa bean shell (CBS) (used alone or within an ice cream) was shown to upregulate nuclear factor erythroid 2 p45-related factor 2 (Nrf2) expression [70]. Nrf2 is a crucial transcription factor protecting cell response against redox stressors and being modulated by oxysterol [71]. Release of interleukin-8 (IL-8) and monocyte chemoattractant protein-1 (MCP-1) induced by oxysterol mixture have been reduced by both CBS and CBS ice cream, confirming the anti-inflammatory potential of cocoa [70]. Moreover, CBS and CBS ice cream have prevented the decrease of tight junction (TJ) protein levels (Claudin 1, occludin, and JAM-A), which are involved in mucosa permeability. Indeed, according to Buckley and Turner, “TJ is a selective permeable barrier which represents the rate-limiting step of paracellular transport” [69]. Polyphenols (such as epicatechin) and methylxanthines are thought to be responsible for these beneficial actions. Although theobromine did not affect the expression of Nrf2 gene, it prevented IL-8, MCP-1, TJ changes and inhibited apoptosis by restoring Bax/Bcl XL proteins in CaCo-2 cells treated with oxysterol mixture [70,72].

In Zucker diabetic fatty rats, cocoa supplementation not only prevented the decrease of TJ Zonula occludens-1 protein caused by diabetic conditions but also presented a higher level than the control group. In this study, cocoa has also demonstrated anti-inflammatory actions by decreasing the level of pro-inflammatory cytokines such as tumor necrosis factor alpha (TNF-α), IL-6 and MCP-1 in the colon [73]. In another study, theobromine has also inhibited the production on TNF-α and MCP-1 by reducing the level of their inducer IL-1β. The reduction of IL-1β could be attributed to the ability of theobromine to suppress cellular NF-κB activation by inhibiting the IκBα activation, nuclear p65 accumulation and promoter activation [74]. In a recent study, procyanidins have modulated 150 gene expressions. For instance, cocoa procyanidin has downregulated the gene KEGG JAK-STAT pathway which induces the expression of the pro-inflammatory IL-12, IL-23, IL-6, IL-28, IL-29 as well as JAK2 and STAT4 [75].

One of the most important threats to the intestinal system is related to dietary consumption. Some foods may be more or less tolerated by our organism and may cause immune response, such as Celiac Disease (CD). Indeed, gliadin is a cereal protein enriched in glutamine and proline residues which, once incompletely digested, are converted into peptide 31–43 (pep 31–43) and peptide 57–68 (pep 57–68). Once they are deaminated by the enzyme tissue transglutaminase-2 (TG2), these peptides become immunotoxic peptides recognized by human leukocytes antigen (HLA)-DQ2 or HLA-DQ8 [76]. They induce the activation of CD4^+^ T cells which produces TG2-specific antibodies and gliadin-specific IFN-γ [77], the most potent inducer of TG2. In a recent study, procyanidin B2 from cocoa has shown to reduce the levels of TG2 induced by IFN-γ or pep 31–43 in Caco-2 cells from 44 to 77%. At a mean concentration of 90 μg/mL, methylxanthine also showed a reduction of TG2 induced by IFN-γ or pep 31–43 by 68% and 62 % for caffeine and 62% and 47% for theobromine [78]. Moreover, one crucial marker of CD is the rise of IL-15, which can induce as well TG2 production. Cocoa B2 has reduced the proinflammatory cytokine IL-15, IL-1β, IL-6, IL-8 induced by IFN-γ or pep 31–43 [78].

During food allergy, allergens engender the differentiation of T cell into T-helper 2 cells which can produce cytokines (IL-4, IL-5, IL-10, and IL-13), resulting of the production of Immunoglobulin E (IgE) antibodies by B cells [79]. IgE is recognized by mast cell receptors named FcεRI and induces the release of histamine, protease and cytokines that revealed anaphylactic response. Polyphenols from cocoa, especially flavonoids, upregulated the gene expression of IgE receptor FcεRI and decrease in rat mast cell protease II (RMCP-II) levels. This marker of fast degranulation would be the consequence of the cocoa-inducing lower rate of IgE. Interestingly, the nature of polyphenol contained in the sample could induce different mechanisms. Indeed, even if both cocoa samples inhibited IL-5 and IL-13 that promote Th2 response, only fermented cocoa inhibits the IL-4 release, which acts in the synthesis of IgE. This explained why fermented cocoa prevented the synthesis of anti-ovalbumin IgE, whereas unfermented cocoa decreased these antibodies only after the end of the diet. Moreover, cocoa intake reduced the release of Th1-cytokines IL-1α, IL-1β and IFN-γ from mesenteric lymph nodes cells and specific IgG antibodies associated with Th2-immune response [80].

To go further, abnormal level in lipid and glucose are associated with inflammation process that has occurred in adipose tissue. Activation of the TLR4 receptor by free fatty acid can trigger c-Jun N-terminal kinase (JNK) and induce a crucial inflammatory transcription factor called NF-κB. Pro-inflammatory cytokine genes and protein levels such as TNF-α and IL-6 decreased with cocoa powder and cocoa extracts in white adipose tissue, suggesting an anti-inflammatory action of cocoa [7].

### 5.2. Cocoa and Obesity

The World Health Organization (WHO) has recognized obesity and overweightness as factors that cause noncommunicable diseases. Two types of adipose tissues (which stores triglycerides and release free fatty acids) can be distinguished: white adipose tissue (WAT) and brown adipose tissue (BAT) [81]. Obesity is thought to be responsible for the inflammation of WAT, leading to lipolysis and an increase in free-fatty acids from triglycerides. Free-fatty acids (FFA) are derived of triglycerides by cleavage of ester bond caused by lipases [82]. They are released from expanding mass fat and are associated with obese human subjects [83]. An increase in FFA level results in the release of inflammatory markers such as TNF-α, IL-6 and IL-10 and the release of ROS [84]. In the long-term, obesity leads to a set of chronic pathologies, including hypertension, type 2 diabetes, and cardiovascular disease [84].

Cocoa has already shown anti-obesity properties by acting in various ways. Obesity is characterized by abnormal fat and glucose levels. After daily intake of flavonoid-rich cocoa products, total blood cholesterol, triglycerides, and LDL-cholesterol were significantly decreased and blood pressure was improved [7,85]. Moreover, (−)-epicatechin, cocoa powder and cocoa extract have caused a significant decrease in body weight gain, fat mass accumulation, dyslipidemia, hyperglycemia, and insulin resistance induced by a high-fat diet intake [7].

A strategy to reduce adiposity and dyslipidemia is based on the upregulation of peroxisome proliferator-activated receptor (PPARγ), peroxisome proliferator-activated receptor γ coactivator 1 α (PGC1α), and Sirtuin 1 (SIRT1). The PPARγ ligand plays a key role because of its ability of lower plasma triglyceride levels and increase HDL-cholesterol [86]. Cocoa enhanced the expression of PPARγ levels related to the high-fat group, in association with a diminution of serum triglycerides and FFA levels [87]. The latter are PGC1α, which can bind to and co-activate PPARγ to contribute to the transports and use of fatty acids [88]. PGC1α could interact with SIRT1 and they regulated energy homeostasis and increase energy expenditure over energy storage in WAT. Cocoa powder, cocoa extract and epicatechin modulated the expression of PGC1α, SIRT1 and PPARγ in WAT, suggesting that one of the possible mechanisms involved in body weight gain and fat mass accumulation would be to increase energy expenditure in the case of hypercaloric diet-induced [7]. PPARγ and C/EBPα act as transcription factors in adipogenesis. In a recent study, theobromine has suppressed the accumulation of lipids and the differentiation of adipocytes by decreasing the expression of PPARγ and C/EBPα (involved in the latter stage of adipogenesis) and by degrading the C/EBPβ (involved in the early stage of adipogenesis) [89,90].

Another strategy is based on the AMP-activated protein kinase (AMPK) pathway. After consuming cocoa in high-fat diet, the expression of AMPK was superior to that of the standard group, suggesting that the AMPK pathway could be activated by cocoa [87]. This enzyme is related to the promotion of β-oxidation, lipogenesis of FFA and lipogenic and fatty oxidation enzymes activities. Indeed, in another study, cocoa polyphenols supplementation activated again the AMPK pathway phosphorylation but also downregulated the expression of Acaca (Acetyl-coenzyme A carboxylase alpha) and Mcat (Malonyl coa ACP acyltransferase) genes in the liver, and Fasn (Fatty acid synthase) and Scd1 (Stearoyl-coenzyme A desaturase) genes in the WAT [91]. They are key enzymes in the fatty acid synthesis and would be regulated by the AMPK pathway. For example, the activation of AMPK reduces Mcat activity and malonyl-coA level that promotes CPT1 (Carnitine palmitoyltransferase 1) activity, resulting in an increase in FFA transport into mitochondria for β-oxydation [92]. Methylxanthines also inhibited the differentiation of preadipocytes via the activation of AMPK and inhibition of the JNK pathway [89]. Whether in the liver or WAT, AMPK pathway modulation could act on various key regulators as PPARα (liver) or PPARγ (WAT) to reduce body weight, fat mass, lipogenesis and increase lipolysis [91]. Lipolysis pathway, which releases FFA and glycerol, could be promoted by methylxanthines by antagonism of adenosine receptors and by enhancing the effects of catecholamines. For instance, lipolysis is managed by the affinity between catecholamines and α or β adrenergic receptors. Caffeine has stimulated lipolysis by modulating the release of noradrenalin and by activating β-adrenergic receptors in the nervous system [93].

Another important pathway is based on the existence of leptin, which is a hormone secreted by WAT and has a key role in regulating body weight. This is due to its ability to influence satiety and energy expenditure. Hyperleptinemia is present in obese humans and is thought to engender a leptin-resistance resulting from a suppressed action in WAT metabolic function. Cocoa revealed a positive correlation between serum leptin levels body weight gain/fat mass accumulation for treatment with cocoa extracts, cocoa powder and epicatechin in high-fat diet. These treatments have also been associated with high downregulation of leptin gene expression in WAT, suggesting that cocoa would reduce hypercaloric diet induced-leptin resistance in WAT [7]. Flavanol-rich powder supplementation in athletes also showed a decrease in leptin levels and a reduction in fat body mass [94].

### 5.3. Cocoa and Diabetes

Diabetes is a set of metabolic diseases characterized by sustained hyperglycemia caused by dysfunctions in insulin metabolism. Nowadays, several diabetes types were well-known, the type-1 diabetes (T1D) resulting from a defect in insulin secretion (accounted by 5–10% of those with diabetes) and the type-2 diabetes (T2D) resulting from a defect in insulin action (90–95%). T1D autoimmune cell-mediated destruction of pancreatic β-cells (by T-cells) are responsible for normoglycemia by stimulating insulin secretion. This can be diagnosed by the presence of some markers such as islet cell autoantibodies, autoantibodies to insulin, autoantibodies to glutamic acid decarboxylase (GAD65), and autoantibodies to the tyrosine phosphatases IA-2 and IA-2β. T2D is characterized by a transient condition called pre-diabetes and a decrease in glucose tolerance/insulin sensitivity that causes abnormal glucose homeostasis [95]. Other specific types of diabetes also exist such as genetic abnormalities of β-cell, genetic defects in the action of insulin, diseases of the exocrine pancreas or endocrinopathies.

In the long term, diabetes can cause many disorders: from retinopathy to vision loss, from nephropathy to renal dysfunction, from peripheral neuropathy to amputations and Charcot joints, from autonomic neuropathy to gastro-intestinal and cardiovascular failures [95]. On a long-term human study (1975–2000), an inverse association between chocolate consumption and diabetes was noted, suggesting as well that chocolate would be beneficial in lower risk for diabetes mellitus [8]. Cocoa is a source of (-)-epicatechin, procyanidins and methylxanthines that have already shown antidiabetic actions [96,97]. In in vitro and in vivo studies, theobromine has been revealed to prevent extracellular matrix (ECM) accumulation of kidney in diabetic models. Indeed, SIRT-1 reduction is one of the characteristic markers of ECM accumulation and can bring diabetic nephropathy (a microvascular disorder that leads to chronic renal dysfunction). SIRT-1 is an NAD^+^-dependent protein of the family of deacetylase, so its activity is linked with NAD+ (nicotinamide adenine dinucleotide plus) levels. Higher level of ROS upholds the activation of NADPH oxidase 4 which is linked with the activation of NAD-dependent DNA repair enzyme, poly(ADP-ribose) polymerase-1 (PARP-1) and (NOX-4), resulting in a depletion of NAD^+^, which cause downregulation of SIRT-1. Theobromine may have increased the SIRT-1 level by reducing the activation of NOX-4 which have blocked the PARP-1 activation and may have restored the NAD^+^ levels [97]. This suggests that theobromine ability to interfere in the chain of events could be a therapeutic strategy in diabetic nephropathy.

A potential antidiabetic action of cocoa could result from a protective action on β-cells against death-inducing factors. Indeed, Fernández-Millán and collaborators have shown that cocoa-rich diet induced an antioxidative action (mainly glutathione peroxidase) that could protect β-cell against oxidative injuries induced by pre-diabetic conditions. As a result, they suggested that cocoa polyphenols would reduce apoptosis of β-cell mass and delay the progression of T2D [98]. This prediabetic phase characterized by insulin resistance and glucose tolerance and its progression could be delayed by changes in lifestyle or treatments. Sucrose or fructose-rich diet could enhance endocrine-metabolic disturbance in rats and induced a prediabetes. A daily uptake of polyphenol-enriched cocoa extract has prevented the increase in homeostasis model assessment-insulin resistance index (HOMA-IR index) in sucrose-rich diet for rats, suggesting that polyphenols of cocoa have a protective effect on insulin resistance [99]. In this study, polyphenols also induced the decrease of P-Akt/Akt and P-eNOS/eNOS ratios. In the literature, the Akt-dependent phosphorylation of Ser is involved in the activation of nitric oxide synthesis by eNOS. Insulin would be involved in Akt-modulated mechanism and would be able to release oxide nitric [100]. Changes in the previous ratios would show that cocoa would be able to prevent insulin resistance. Moreover, cocoa showed an involvement in improving insulin resistance by decreasing insulin resistance receptor phosphorylation (IRS-1, Ser 307, and Ser 636/639) and activating the Glycogen Synthase Kinase 3/Glycogen Synthase (GSK3/GS) pathway. Indeed, this could be a good antidiabetic strategy because GSK3-β is a capital substrate of the phosphatidylinositol 3-kinase/protein kinase B (PI3K/AKT) signaling involved regulating glycogen synthesis, and GS plays an important role in insulin resistance [101].

Hyperglycemia may be managed by the ability of cocoa (−) epicatechin and other flavonoids to increase GLUT-2 levels [102]. GLUTs maintain the inter- and extracellular balance of glucose and may be affected in diabetes state. Although Bowser and collaborators have inhibited GLUT-4 activity, a significant reduction in glucose uptake in human skeletal muscle was noted in the presence of cocoa extract and procyanidin fractions. This behavior suggested that GLUT-4 activity would be involved in the antidiabetic mechanism cocoa extracts [103]. A cocoa-rich diet has also induced a reduction in postprandial plasma glucose elevation. This is believed to be due to increased early insulin secretion and the activity of the glucagon-like peptide-1(GLP-1) activity [104]. These antidiabetic actions would not be visible for all cocoa polyphenols because only the catechin cocoa fraction showed an enhancement in glucose-stimulated insulin secretion, whereas treatments with crude cocoa extract or procyanidin fraction showed no improvement [4].

Cordero-Herrera and collaborators indicated that in the Zucker diabetic fatty model, an increase and a decrease in phosphoenolpyruvate carbokinase (PEPCK) and Glucokinase (GK) levels, respectively, are observed. Their cocoa-rich diet modulated these gluconeogenic and glycolytic enzymes contents by thwarting the increase of PEPCK and boosted the GK contents in liver [105]. Negatively regulated by phosphorylation, GS activity is characterized by a decrease in the ratio of phospho/total glycogen synthase. This decrease was noted for all treatments with crude cocoa extract, monomers, and polymer procyanidin fractions, suggesting that cocoa would modulate GS activity [103].

In addition to modulating various parameters, cocoa crude extract and procyanidin fractions suggested an ability to mimic the action of insulin in skeletal muscle in absence of insulin. These fractions increased glucose uptake, glycogen synthesis by AKT pathway, GS activity and GLUT-4 activity in absence of insulin [103]. These data suggested a promising double opportunity in the treatment of diabetes. Indeed, for T1D where insulin is deficient, cocoa polyphenol could mimic its activity, whereas in T2D, where insulin is less effective, cocoa also showed insulin modulation activities. To summarize health benefits of cocoa polyphenols and methylxanthines on dietary metabolism disorders, the Table 2 presents the impact and the metabolic pathway involved.

### 5.4. Cocoa and Gut Microbiota

Gut microbiota (GM) is a set of microorganisms which is in the digestive tract named intestinal microbiome. Despite a huge diversity in bacteria, all of them belong to four bacterial phyla: *Firmicutes* (64%), *Bacteroidetes* (23%), *Proteobacteria* (8%) and *Actinobacteria* (3%) [106]. However, composition of the microbiota is related to various factors—such as age, genetic and geographical origins, health, lifestyle, and the way the individual was born [107]—which highlight that each human has its own unique microbiota profile. One of the most crucial functions of GM is to occupy intestinal surfaces and consequently protect them against a colonization by harmful bacteria species such as *E.coli*.

Even if cocoa has already shown antimicrobial activities, it could be appropriated to maintain the integrity of GM while it also could engender the emergence of new microorganism. Indeed, the daily consumption of rich flavanol cocoa drink increased the growth of *Lactobacillus* spp. and *Bifidobacterium* spp. and decreased the *clostridium pefringens* in comparison with a control group. This modulation of cocoa presented benefits because *Lactobacillus* spp. and *Bifidobacterium* spp prevent the apparition of pathogenic species and *Clostridium pefringens* is involved in colonic cancer and onset of inflammatory bowel disease [108].

In cocoa-enriched diet, some compounds could influence the emergence of new bacteria species of GM. Indeed, Martin-Pelaez and collaborators noted that a cocoa diet containing 0.4% polyphenols and 0.25% theobromine induced the emergence of seven fecal-detected bacteria species belonging to the *Actinobacteria*, *Cyanobacteria*, *Firmicutes,* and *Proteobacteria* phyla. However, consumed without polyphenols, the cocoa-diet containing 0.25% theobromine was the only one that allowed the emergence of *Candidatus Arthromitus* (*Firmicutes* phylum). In addition, theobromine was the only one that reduced the counts of *Bifidobacterium* spp., and *Firmicutes* phylum (*Streptococcus* spp., and *C. histolyticum–C. perfringens)* [109]. This suggests possible interferences in activities between several cocoa compounds. Indeed, 0.25% theobromine diet increased almost fourfold *Tenericutes* phylum in gut microbia, whereas cocoa-diet did not, suggesting the possible delay interference of others cocoa compounds [109]. An interesting hypothesis is based on the possibility that methylxanthines and polyphenols may have opposite actions. Indeed, theobromine, taken individually, induced the reduction of Ruminococcus flavefaciens, (cellulolytic bacteria) and the increase of Erysipelotrichaceae while flavonoids induced opposite actions [109,110,111].

The ability to modulate qualitatively and quantitatively the microbiota plays a key role in health because several species would be responsible for pharmacological activities. Indeed, cocoa has revealed to be a prebiotic, “a substrate that is selectively utilized by host microorganism and conferring a health benefit” [112]. As we explained, cocoa polyphenols (flavanols) induced an increase in *Lactobacilli* and *Bifidobacteria* [113]. These species influenced the differentiation of the regulatory T cells, a substantial source of Il-10, an anti-inflammatory cytokine which presents anti-obesity actions by suppressing adipocyte energy expenditure, thermogenesis and by reducing insulin resistance [114]. Consequently, as prebiotics, cocoa flavanols might have immunomodulatory effects by interfering with GM.

Gastro-intestinal disorders have been correlated with the modulation of microbiota composition. The ability of polyphenols of cocoa to increase *Lactobacillus* would explain why this behavior was observed after dark chocolate consumption of obese volunteers. The abundance of *Lactobacillus* would be associated with a reduction in liver associated blood markers of oxidative damage and inflammation [115]. Today, it is well-known that microbiota work in symbiosis with the host and have an influence on food allergy by inducing oral tolerance. This is possible because GM stimulates steadily GALT by presenting various antigens which induce in long term mucosa immune tolerance [116]. In an oral sensitized model with rats, a 10% cocoa diet modified the GM composition by reducing *F**irmicutes* and *Proteobacteria* phyla and by increasing *Tenericutes* and *Cyanobacteria* phyla. Abundance of *Proteobacteria* phylum was positively correlated with a high level of intestinal Immunoglubulin A (IgA) [117] because *Proteobacteria* phylum are thought to be the main inducer of IgA by B-cells [10]. Camps-Bossacoma and collaborators have observed that cocoa induced a stabilization on IgA level for oral-sensitized animal, whereas both groups not fed with cocoa (whether they are oral-sensitized or not) presented a constant increase in IgA during the study. This behavior suggested that polyphenols and methylxanthines of cocoa have tolerogenic action by influencing GM (especially *Proteobacteria* phylum) correlating with intestinal IgA level.

Diabetic rats presented a different GM than healthy group characterized by a higher *Proteobacteria* (3.6-fold higher), *Tenericutes* (2.8-fold), *Actinobacteria* (2.6-fold) phyla and a lower *Verrucomicrobia* (by 76.9%) phylum counts [73]. Polyphenols showed an influence by reducing *Proteobacteria* abundance induced by diabetes [73,118]. Cocoa polyphenols have also significantly induced modulation in *Firmicutes* (1.4-fold increase) and *Deferribacteres* phyla (9.3-fold increase) and decreased the relative abundance of *Cyanobacteres* phylum (by 74.9%) compared to diabetic group [73]. This suggested that cocoa interferes qualitatively and quantitatively with the GM where some species have more or less efficiency on various health benefits. To valued cocoa potential in GM, some innovations are developed such as sucrose-free milk chocolate used as a prebiotic or also chocolate used as carriers of encapsulated probiotics [119,120]. Both formulations expressed significant survival rate of probiotics during the in vitro gastrointestinal digestion [119]. Figure 3 presents the influence on GM species and provides examples of the relationships between polyphenols and methylxanthines on dietary metabolic disorders. Indeed, as mentioned, the emergence of *Lactobacilli* and *Bifidobacteria* has caused a decrease in Treg cells, which is a substantial source of IL-10. This cytokine is involved in insulin resistance. Moreover, without the production of IL-10, white adipocyte could be converted into “beige adipocyte”. The latter could increase the energy expenditure by UCP1 (Uncoupling protein 1) and lead to weight loss [98]. Methylxanthines have shown to decrease the level of *Clostridium pefringens* which could induce a prevention of colon cancer and a delay in the development of inflammatory bowel disease [92].

## 6. Conclusions

Cocoa has many varieties that can distinguished by morpho-geographical and genetic factors. These factors influence the composition of each part of the plant, including cocoa beans. In recent studies, two groups of compounds were of great interest, including polyphenols and methylxanthines. As mentioned, these compounds are thought to be responsible for many health benefits, particularly antioxidant and anti-inflammatory capacities in the digestive tract. Indeed, this review focused on the influence of polyphenols and methylxanthines on dietary metabolic disorders as diabetes, obesity, and intestinal inflammations, which are constantly increasing. Our work has shown that all these disorders are related and can interfere with each other, especially with the gut microbiota (GM). Recently, several studies have presented the GM contribution in metabolic disorders. The Dabke and collaborators model suggest that a high-fat, low-fiber diet could lead to dysbiosis (unbalanced GM conditions) which could lead to altered TJ integrity. The dysfunction of the permeability of intestinal epithelial could lead to chronic inflammation of liver and adipose tissue by certain threats to the body (LPS, metabolites…). As a result of insulin resistance, increased FFA levels and dyslipidemia, this chain of response could result in a high risk of obesity, diabetes, and cardiovascular diseases [125]. This was supported by the fact that infusion of GM from lean volunteers to subjects with metabolic disorders demonstrated an increase in insulin sensitivity [126].

In this paper, we discussed the dual action of cocoa compounds (internal and external) on dietary metabolic disorders and GM. Cocoa polyphenols and methylxanthines modulated the GM itself leading to the emergence of 11 new bacterial species such as *Bacteroides*, *Firmicutes* and *Proteobacteria* phyla. In addition, an external action also exists because these compounds have affected more than the gut. For example, the influence of these compounds on the abundance of *Lactobacilli* and *Bifidobacteria* phyla leads to an increase in pro-inflammatory cytokine IL-10 levels. This rise could delay obesity and body weight by increasing thermogenesis and increasing insulin-sensitivity, a key parameter of diabetes. Our work has also shown that polyphenol and methylxanthine in cocoa can have opposites actions.

This suggests that cocoa is a complex matrix, and its composition is the masterpiece of its effectiveness. Indeed, it is well-known that not all the native compounds reach the bloodstream. Only a few compounds, especially small ones such as (−)-epicatechin, can surround the entire digestive process. Thus, all parameters influencing the composition, such as variety, geographic origins and fermentation process, must be managed. Knowing that the latter engenders a loss of 90% of (−)-epicatechin content, one may wonder if alternatives could be found, such as raw chocolate that is made with unfermented and/or unroasted cocoa beans. Even if the raw material is crucial, the extraction procedure is not negligible because the choice of extraction solvent or/and the extraction method could play a key role. For example, 70% acetone results in a higher total polyphenol content in cocoa pod husk extract than 70% ethanol (94.92 and 49.92 mg GAE/g extract, respectively). Water and ethanol provide a partial extraction of oligomeric polyphenols, but high polymers are not extracted at all [5,127]. Moreover, ultrasound (2h-60Hz) provided extracts with higher total polyphenol content than those obtained by agitation (6h-200rpm) [128].

Cocoa can be considered a “Swiss knife” because it offers many possibilities for use in various forms in the agri-food, cosmetic and other sectors. For instance, the cocoa shell, incorporated into ice cream, showed prebiotic activities, while the cocoa incorporated in mouth products showed anticariogenic and antibacterial capacities [70,129]. As a conclusion, further studies are needed to fully characterize the potential actions of cocoa compounds on dietary metabolic disorders and GM.

## Figures and Tables

**Figure 1 foods-10-02049-f001:**
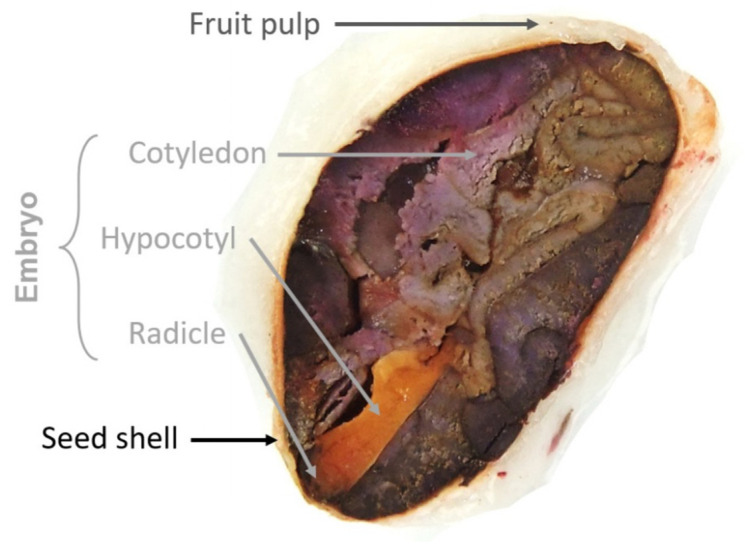
Morphology of a cocoa bean adapted from Kadow, 2020 [24].

**Figure 2 foods-10-02049-f002:**
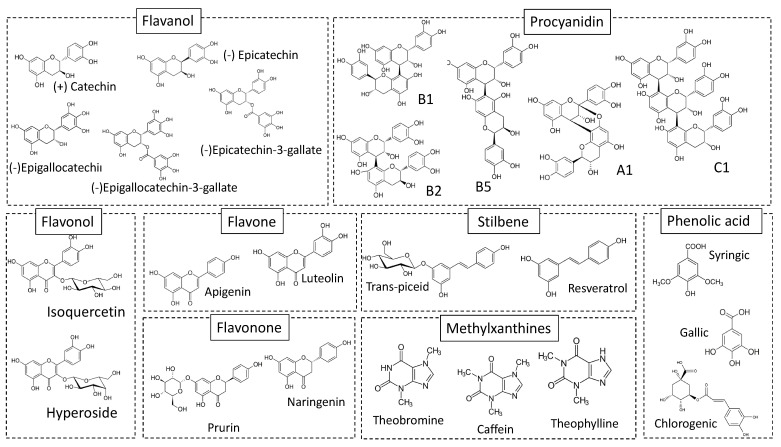
Polyphenols and methylxanthines structures occurring in cocoa [5,28].

**Figure 3 foods-10-02049-f003:**
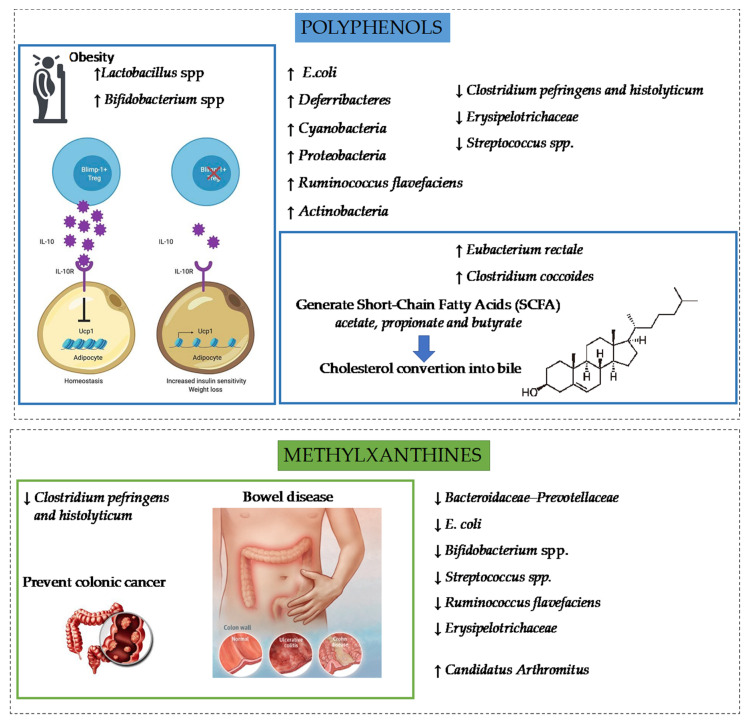
Benefits of polyphenols and methylxanthines on gut microbiota some influence on dietary disorders related to them. ↑ mean an increase in the microorganism content ↓ mean a decrease in the microorganism content [73,109,111,113,114,121,122,123,124].

**Table 1 foods-10-02049-t001:** Cocoa almond and shell compositions (% of dry matter) [23,26,27].

	Almond (%)	Shell (%)
Moisture	2–50	4–11
Fat	48–57	2–6
Protein	10–22	13–20
Starch	6–9	6.5–9
Fiber	2–3.2	13–19
Polyphenol	14–20	-
Theobromine	0.8–2.7	0.2–1.3
Caffeine	0.1–0.8	0.04–0.6
Ash	2.6–4.2	6.5–20.7

**Table 2 foods-10-02049-t002:** Impacts of cocoa compounds on dietary metabolic disorders by and identification of metabolic pathway involved.

	Impact of Polyphenols or/and Methylxanthines from Cocoa	Metabolic Pathway Involved	Sources
Intestinal inflammation	Protection against redox stressors, Prevention of intestinal tigh junction, Reduction of inflammation caused by oxysterols,	Upregulation of Nrf2 expression, Decrease of IL-8 and MCP-1 levels, Restoring Bax/Bcl XL proteins, Avoid decrease of Claudin 1, occludin, and JAM-A levels	[70,72]
	Reduction of inflammation in colon	Decrease of TNF-α, IL-6 and MCP-1 levels	[73]
	Reduction of inflammation	Induce cellular NF-κB activation by inhibiting the IκBα activation, nuclear p65 accumulation and promoter activation	[74]
	Reduction of inflammatory markers such as IL-12, IL-23, IL-6, IL-28, IL-29, JAK2 and STAT4	Modulation of 150 gene expressions, downregulation of gene KEGG JAK-STAT pathway	[75]
	Reduction of celiac disease markers	Decrease of the levels of TG2, IL-15, IL-1β, IL-6, IL-8 induced by IFN-γ or pep 31–43	[78]
	Reduction of prevention of food allergy IgE	Upregulation of the gene expression of IgE receptor FcεRI, decrease of rat mast cell protease II (RMCP-II) levels, prevention of the synthesis and decrease of anti-ovalbumin IgE,	[80]
Obesity	Reduction of adiposity, dyslipidemia, plasma triglyceride levels and increase HDL-cholesterol	Upregulation of PPARγ, PGC1α and SIRT1	[86,87,88]
	Reduction in lipid accumulation and inhibition of the differentiation of preadipocytes	Decrease the expression of PPARγ and C/EBPα and C/EBPβ, activation of AMPK and inhibition of the JNK pathway	[89,90]
	Promotion of β-oxidation, lipogenesis of FFA and activation of lipogenic and fatty oxidation enzymes	Activation of AMPK pathway, Down-regulation of the expression of Acaca Mcat, Fasn and Scd1 genes	[87,91]
	Improve lipolysis pathway and release of FFA and glycerol	Antagonism potential of adenosine receptors, Enhance the effects of catecholamines, Modulation of the release of noradrenalin, Activation of β-adrenergic receptors	[93]
	Influence satiety and energy expenditure, reduce hypercaloric diet induced-leptin resistance, activation of fat body loss	Down regulation of leptin gene expression	[94]
Diabetes	Prevention against ECM accumulation of kidney	Increase of SIRT-1 level by reducing activation of NOX-4 which have blocked the PARP-1 activation and restoration of NAD^+^ levels	[97]
	Protection against death-inducing factors on β-cells, Reduction of apoptosis of β-cell mass and delay on the progression of T2D	Antioxidative actions (mainly glutathione peroxidase)	[98]
	Increase of HOMA-IR index	Decrease of P-Akt/Akt and P-eNOS/eNOS ratios, decrease of insulin resistance receptor phosphorylation (IRS-1, Ser 307, and Ser 636/639) and activation of the GSK3/GS pathway	[99,101]
	Modulation of glucose metabolism with or without insulin	Activation of GLUT-4 activities, Increase of GLP-1 activity, Increase of PEPCK and GK levels,	[103,104,105]
	Reduction in postprandial plasma glucose elevation.	Increase insulin secretion and activation of GLP-1 activity	[104]

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
