# Peer review of "Benefits of Polyphenols and Methylxanthines from Cocoa Beans on Dietary Metabolic Disorders"

_foods, 2021, doi:10.3390/foods10092049_

Round 1

Reviewer 1 Report

The Review entitled “Benefits of polyphenols and methylxanthines from cocoa beans on dietary metabolic disorders” by Jean-Marie et al, is an interesting overview of the beneficial effect of cocoa beans on metabolic disorders such as diabetes and obesity.

The review is well organized describing the biological targets of the bioactive molecules found in cocoa.

Minor suggestions:

  1. The chemical name of catechin should be revised inserting the position of gallate functionalization (for example Epigallocatechin-3-gallate)
  2. Figures showing the beneficial effects of polyphenol and methylxanthines should be inserting to help the readers in understanding the inhibition and/or activation of the signalling pathways described.
  3. The anti-inflammatory properties of cocoa beans is also related to the antioxidant activity of bioactive molecules (doi: 10.1002/mnfr.201000412; doi: 10.3390/nu8040212). This point should be better described.

Author Response

The Review entitled “Benefits of polyphenols and methylxanthines from cocoa beans on dietary metabolic disorders” by Jean-Marie et al, is an interesting overview of the beneficial effect of cocoa beans on metabolic disorders such as diabetes and obesity.The review is well organized describing the biological targets of the bioactive molecules found in cocoa.

Response: Thank you very much for your comments which helped us improve this manuscript. The entire document submitted an English revision. The lines mentioned in this document refer to the line number in the review in “tracking change” mode.

Minor suggestions:

  1. The chemical name of catechin should be revised inserting the position of gallate functionalization (for example Epigallocatechin-3-gallate)

Response : Thanks for your kind reminders. Clarification was made on the figure 2

2. Figures showing the beneficial effects of polyphenol and methylxanthines should be inserting to help the readers in understanding the inhibition and/or activation of the signalling pathways described.

Response : Thanks for your suggestion. We add this part of explanation in lines 794-803

 “The Figure 3 presents the influence on GM species and provides examples of the relationships between polyphenols and methylxanthines on dietary metabolic disorders. Indeed, as mentioned, the emergence of Lactobacilli and Bifidobacteria has caused a decrease in Treg cells, which is a substantial source of IL-10. This cytokine is involved in insulin resistance. Moreover, without the production of IL-10 white adipocyte could be convert into “beige adipocyte”. The latter could increase the energy expenditure by UCP1 (Uncoupling protein 1) and lead to weight loss [98]. Methylxanthines have shown to decrease the level of Clostridium pefringens which could induce a prevention on colon cancer and a delay on the development of inflammatory bowel disease [92].“

3. The anti-inflammatory properties of cocoa beans is also related to the antioxidant activity of bioactive molecules (doi: 10.1002/mnfr.201000412; doi: 10.3390/nu8040212). This point should be better described.

Response : Thanks for the documentation provided. We insert some discussion of the related antioxidant activity in the section below (lines 375-415): “If inflammation is a beneficial and necessary process, when it becomes chronic and able to attack its own host, it becomes a problem. One of its disadvantages is the fact that inflammation and oxidative stress are two closely related phenomena. Indeed, oxidative stress (condition of imbalance between the production and the destruction of oxidative compounds that become toxics to cells) may be a consequence of inflammation. Macrophages and polymorphonuclear neutrophils are important sources of oxidizing compounds called reactive oxygen species (ROS) through a membrane enzyme called NADPH oxidase that is activated after a stress trigger by inflammatory process. To protect their integrity, the cells induced the release of proteolytic enzymes that produce superoxide anion O2-•. This free radical could lead to the production of other toxic radicals as hydrogen peroxide H2O2 (after dismutation), hypochlorous acid HClO, hydroxyl radical OH and nitric oxide NO [64]. Paradoxically, oxidative stress may be also the cause of inflammation. Indeed, H2O2 and OH can lead to the production of proinflammatory cytokines such as IL-6 and TNF-α by the modulation of NFκB, while they are themselves modulated by this factor [65,66]. There was a correlation between antioxidant potential of polyphenols and the molecular effects of anti-inflammatory and chemo-protective activities. This may be due to their ability to modulate signaling cascades and gene expressions involved in cell-proliferation, apoptosis and chronic inflammation [67], but also to their ability to scavenge free radicals [68].”

Reviewer 2 Report

The paper of Elodie et al. is a clear resume of beneficial effects on dietary metabolic disorders, produced by polyphenols and methylxanthines contained in Theobroma cacao L.. This piece of work is well written and organized, it represents an important source of information concerning metabolic pathways and gut microbiota modification triggered by cocoa-derived compounds.

Nevertheless, it is opinion of this referee that few minor modifications/additions are required.

Some comments are listed below.

Line 9: Please, correct writing form: Theobroma cacao L.

Line 22: Please, add Linneus abbr. to the Theobroma cacao.

Line 38,39: Theo and Broma should be written in italics.

Line 39: Please, add Linneus abbr. to the Theobroma cacao.

Line 71: A scheme to depict the Cocoa beans structure could be useful for readers.

Line 88: In “Bean Composition” this referee suggests to briefly list the main analytical methods applied to determine polyphenols/methylxanthines fractions in cocoa. This is due to the central role played by these components in the paper. For this purpose, one possibility is to exploit Data in Brief publications, such as:

Grillo, G.; Boffa, L.; Binello, A.; Mantegna, S.; Cravotto, G.; Chemat, F.; Dizhbite, T.; Lauberte, L.; Telysheva, G., Analytical dataset of Ecuadorian cocoa shells and beans, Data in Brief, 2019, 22, 56-64. doi:10.1016/j.dib.2018.11.129.

Line 521-525: In relation with the several forms in which cocoa can be exploited, listed in the last part of the conclusions, it is advisable to mention the existence of dedicated extraction procedures. The latter can recover and concentrate bioactive fractions from raw cocoa and even from processing residues. An helpful reference can be the following:

Campos-Vega, R.; Nieto-Figueroa, K.H.; Oomah, B.D., Cocoa (Theobroma cacao L.) pod husk: Renewable source of bioactive compounds, Trends in Food Science and Technology, 2018, 81, 172-184. doi: 10.1016/j.tifs.2018.09.022.

Bibliography: Please modify the references according to MDPI guidelines (bolded publication year, italics for Journal abbreviation, issue, etc…)

Final remark:

It is opinion of this referee that, to improve the effectiveness of the whole manuscript, one summarizing table should be implemented in the manuscript. The latter should contain the metabolic pathway and/or the enzymes and effect addressed by cocoa, divided by each dietary disorder. Reporting for each line the relative references would surely enhance the citability of the whole work, valorising the extensive research carried out by the authors. This type of overview is not necessary for gut microbiota, being already clearly depicted in Figure 2.

Author Response

The paper of Elodie et al. is a clear resume of beneficial effects on dietary metabolic disorders, produced by polyphenols and methylxanthines contained in Theobroma cacao L.. This piece of work is well written and organized, it represents an important source of information concerning metabolic pathways and gut microbiota modification triggered by cocoa-derived compounds.

Nevertheless, it is opinion of this referee that few minor modifications/additions are required.

Response: Thank you very much for your comments which helped us improve this manuscript. The entire document submitted an English revision. The lines mentioned in this document refer to the line number in the review in “tracking change” mode.

Some comments are listed below.

Line 9: Please, correct writing form: Theobroma cacao L.

Line 22: Please, add Linneus abbr. to the Theobroma cacao.

Line 38,39: Theo and Broma should be written in italics.

Line 39: Please, add Linneus abbr. to the Theobroma cacao.

Response : Thanks for your kind reminders. Modifications were made.

Line 71: A scheme to depict the Cocoa beans structure could be useful for readers.

Response : Thanks for your suggestion. In this section, we add a figure illustrating the morphology of cocoa bean. (Figure 1-line 91)

Line 88: In “Bean Composition” this referee suggests to briefly list the main analytical methods applied to determine polyphenols/methylxanthines fractions in cocoa. This is due to the central role played by these components in the paper. For this purpose, one possibility is to exploit Data in Brief publications, such as:

Grillo, G.; Boffa, L.; Binello, A.; Mantegna, S.; Cravotto, G.; Chemat, F.; Dizhbite, T.; Lauberte, L.; Telysheva, G., Analytical dataset of Ecuadorian cocoa shells and beans, Data in Brief201922, 56-64. doi:10.1016/j.dib.2018.11.129.

Response: Thanks for your suggestion. We add some analytical methods in lines “200-213” and lines “349-357” for polyphenols and methylxanthines, respectively.

Line 521-525: In relation with the several forms in which cocoa can be exploited, listed in the last part of the conclusions, it is advisable to mention the existence of dedicated extraction procedures. The latter can recover and concentrate bioactive fractions from raw cocoa and even from processing residues. An helpful reference can be the following:

Campos-Vega, R.; Nieto-Figueroa, K.H.; Oomah, B.D., Cocoa (Theobroma cacao L.) pod husk: Renewable source of bioactive compounds, Trends in Food Science and Technology201881, 172-184. doi: 10.1016/j.tifs.2018.09.022.

Response : Thank you for the documentation provided. We modified the conclusion with this section below in lines 864-870:  “Even if the raw material is crucial, the extraction procedure is not negligible because the choice of extraction solvent or/and the extraction method could play a key role. For example, 70% acetone results in a higher total polyphenol content in cocoa pod husk extract than 70% ethanol (94.92 and 49.92 mg GAE/g extract, respectively). Water and ethanol provide a partial extraction of oligomeric polyphenols, but high polymers are not extracted at all [5,125]. Moreover, ultrasound (2h-60Hz) provided extracts with higher total polyphenol content than those obtained by agitation (6h-200rpm) [126].”

Bibliography: Please modify the references according to MDPI guidelines (bolded publication year, italics for Journal abbreviation, issue, etc…)

Response : Thanks for the kind reminder, we modified the references list.

Final remark:

It is opinion of this referee that, to improve the effectiveness of the whole manuscript, one summarizing table should be implemented in the manuscript. The latter should contain the metabolic pathway and/or the enzymes and effect addressed by cocoa, divided by each dietary disorder. Reporting for each line the relative references would surely enhance the citability of the whole work, valorising the extensive research carried out by the authors. This type of overview is not necessary for gut microbiota, being already clearly depicted in Figure 2.

Response : Thanks for your proposition, we made a summarizing table (Table 2-line 688) for improving the reading.

Round 2

Reviewer 1 Report

The manuscripts is now suitable for the publication in Foods